# The Effect of Copper Sulfide Stoichiometric Coefficient and Morphology on Electrochemical Performance

**DOI:** 10.3390/molecules28062487

**Published:** 2023-03-08

**Authors:** Yanhong Ding, Rongpeng Lin, Shuicheng Xiong, Yirong Zhu, Meng Yu, Xiaobo Duan

**Affiliations:** 1College of Materials and Advanced Manufacturing, Hunan University of Technology, Zhuzhou 412008, China; 2GEM Co., Ltd., Shenzhen 518101, China

**Keywords:** copper sulfide, supercapacitor, specific capacity, hydrothermal method, morphology

## Abstract

In this work, CuS, Cu_7_S_4_, Cu_9_S_5_, Cu_7.2_S_4,_ and Cu_2_S with the same morphology were successfully synthesized by the hydrothermal method. According to the calculation, their galvanostatic charge-discharge (GCD) curves were 43.29 (CuS), 86.3 (Cu_7_S_4_), 154 (Cu_9_S_5_), 185.4 (Cu_7.2_S_4_), and 206.9 F/g (Cu_2_S) at the current density of 1 A/g. The results showed that the energy storage capacity of copper sulfide with the same morphology increased with the increase of the copper sulfide stoichiometric coefficient. At the second part of this work, the agglomerated cuprous sulfide and the microporous cuprous sulfide were successfully prepared, respectively. In addition, the porous spherical cuprous sulfide was annealed to get nano cuprous sulfide. It is found that the specific capacity of the agglomerated structure is the highest, which had reached 206.9 F/g at the current density of 1 A/g, and 547.9 F/g at the current density of 10 A/g after activation.

## 1. Introduction

With the rapid development of energy technology, electric energy has become the protagonist of the energy era [1]. To meet the needs of rapid development, more and more high-efficient and environmentally friendly new energy sources and energy-storage devices are needed. Supercapacitors with high-power density, a long cycle life, and a fast-charging capacity have attracted the attention of electrochemical researchers in recent years [2,3]. Element copper ranks in the fourth period and the IB group in the periodic table with atomic No. 29. One electron in the outermost layer makes copper have more valence commonly. Rich raw materials, low price, excellent conductivity and easy machining make copper metal play an important role in industrial production and application [4]. Due to the more complex valence state of the sulfide phase and the higher electrochemical performance, transition metal sulfides as electrode materials for supercapacitors have attracted more attention. A high theoretical capacitance, high power density, abundant valence state, REDOX sites, and a low price make copper metal sulfide a widely concerned energy storage [5,6]. As for copper metal sulfide as electrode materials for supercapacitors, many researchers have reported their work. Wang Xiang et al. [7] used Cu_2_O, synthesized by chemical precipitation, and Cu_2_O as the template via a hydrothermal ion-exchange method to prepare nano Cu_7_S_4_, which provided a specific capacitance of 275 F/g at a current density of 1 A/g. Zhou et al. [8] used an ion exchange reaction to regulate the morphologies of Cu_7.2_S_4_ and synthesized submicron and micron porous spheres of Cu_7.2_S_4_. The material has a specific capacitance of 491.5 F/g at a current density of 1 A/g. Ravindra N prepared Cu_2_S of flower-like nanotubes and integrated nanotubes by a continuous ion-adsorption reaction with a specific capacitance of 761 F/g and 470 F/g at a scanning rate of 5 mV/s [9]. Therefore, via different synthesis methods and a post-treatment of materials, adjusting the structure morphology of copper sulfide and so on contributed to the study of the structure with larger specific capacitance. Copper metal sulfide materials have a huge application potential in the field of supercapacitor materials.

In this work, to better explore the stoichiometric coefficient affect energy storage mechanism of different copper sulfides, CuS, Cu_7_S_4_, Cu_9_S_5_, Cu_7.2_S_4,_ and Cu_2_S with the same morphology were synthesized by hydrothermal method and adjusting morphology variables. The electrochemical properties of different copper sulfides were compared, and the energy storage performance of copper sulfides was discussed. Cuprous sulfide has good thermal stability, which will not decompose into cuprous sulfide and sulfur at high temperatures. Therefore, cuprous sulfide with good thermal stability was also selected as the research object. In order to further explore the effect of morphology and structure on the energy storage capacity of copper sulfide, the morphology of cuprous sulfide was regulated by different synthesis methods so as to carry out an electrochemical test and an exploration on different morphology of cuprous sulfides.

## 2. Results and Discussion

### 2.1. Energy Storage of Different Copper Sulfide Materials

#### 2.1.1. XRD Pattern Analysis

Figure 1a shows the XRD map of the precursor of copper sulfide prepared by Cu(CH_3_COO)_2_·H_2_O and TAA. The lower part is a PDF card of Cu_4_(SO_4_)(OH)_6_ material. Through the comparative analysis of the map, it is found that the 2θ angles located at 11.3°, 13.8°, 16.5°, 22.8°, 27.9°, 33.5°, 35.6°, and 48.3° corresponded to the (110), (200), (210), (220), (400), (420), (22-2), and (250) crystal faces of Cu_4_(SO_4_)(OH)_6,_ respectively. However, in addition to the diffraction peak of Cu_4_(SO_4_)(OH)_6_, the pattern has other diffraction peaks, indicating that the substance is a mixture. It may contain copper, sulfur, hydrogen, and oxygen based on the reactants. The diffraction peaks of the other 2θ of 16.2°, 18.2°, 18.8°, 22.3°, 24.0°, and 31.5° are exactly corresponding to the (−110), (020), (011), (021), (020), and (−221) crystal faces of copper sulfate pentahydrate. By XRD analysis of the precursor, it is proved that the precursor is a mixture of Cu_4_(SO_4_)(OH)_6_ and CuSO_4_·5H_2_O. The specific reaction equation is shown in Equation (1):(1)$$\mathrm{Cu}{\left({\mathrm{CH}}_{3}\mathrm{COO}\right)}_{2}\cdot H_{2}O+{\mathrm{CH}}_{3}{\mathrm{CSNH}}_{2}\to {\mathrm{Cu}}_{4}\left({\mathrm{SO}}_{4}\right){\left(\mathrm{OH}\right)}_{6}+{\mathrm{CuSO}}_{4}\cdot 5H_{2}O\;$$

Cu_x_S_y_ (I, II, III, IV, V; x and y are stoichiometric coefficients) was successfully prepared by the reaction of copper acetate monohydrate with TAA under different conditions. XRD images of the sample are shown in Figure 1b–f. The upper part of the figures with the same color are the test curves of the sample, and the lower part is the PDF standard card of the corresponding material. The intensity peaks of 27°, 29.1°, 31.6°, 32.7°, 47.8°, 52.6°, 59.2°, and 73.9° in Figure 1b were analyzed. It is found that the crystal faces of (101), (102), (103), (006), (110), (108), (116), and (208) of CuS coincided with the standard PDF cards of CuS (PDF#06-0464). Compared with the standard card of Cu_7_S_4_ (PDF#33-0489), the diffraction peaks in Figure 1c are found in 26.4°, 29.7°, 31°, 33.9°, 35.3°, 37.8°, 38.9°, 46.7°, and 48.8° diffraction peaks corresponding to the crystal plane (16,0,0), (804), (18,2,1), (20,0,1), (20,4,0), (155), (10,10,3), (0,16,0), and (886). In Figure 1d, the diffraction peaks at 2θ of 27.8°, 29.2°, 32.2°, 41.5°, 46.2°, and 54.7° are consistent with the crystal faces of Cu_9_S_5_ (PDF#47-1748) at (0,0,15), (107), (1,0,10), (0,1,17), (0,1,20), and (0,1,15). In addition, an analysis of the map in Figure 1e shows that the substance is Cu_7.2_S_4_ (PDF#24-0061) because its strong peak 2θ equals 27.8°, 32.1°, 46.2°, 54.8°, and 67.1°, and corresponds to the (111), (200), (220), (311), and (400) crystal faces of Cu_7.2_S_4_. The (100), (102), (110), (103), (112), and (114) crystal faces of Cu_2_S at 26°, 37.3°, 46°, 48.5°, 54°, and 73.9° of 2θ in Figure 1f correspond to the crystal faces of Cu_2_S.

To analyze XRD figures of five kinds of samples, it is proved that CuS, Cu_7_S_4_, Cu_9_S_5_, Cu_7.2_S_4,_ and Cu_2_S have been successfully prepared by controlling the conditions of hydrothermal synthesis with Cu(CH_3_COO)_2_·H_2_O and TAA as raw materials.

#### 2.1.2. Morphological Analysis

Figure 2a,c,e,g,i are the scanning images of CuS, Cu_7_S_4_, Cu_9_S_5_, Cu_7.2_S_4,_ and Cu_2_S. As shown in Figure 2**,** the morphology of all these five kinds of copper sulfide is agglomerated particles with particle diameters ranging from 10 to 55 μm. The purpose of adjusting the five samples to the same morphology is to avoid the influence of morphology on the following electrochemical test. When only the ratio of copper to sulfur atoms is univariate, the comparability of data from subsequent electrochemical tests can be better guaranteed. It can be seen from Figure 2b,d,f,h,j that the large granularity of copper sulfide is mainly due to the agglomeration of small particles. The main reason is that the viscosity of the solvent is small [10].

At room temperature, TAA was hydrolyzed and combined with Cu^2+^ to form flocculent precursor Cu_4_(SO_4_)(OH)_6_ and copper sulfate pentahydrate. As shown in Equation (2), with the rise of hydrothermal temperature, TAA continues to hydrolyze to form S^2−^. Then S^2−^ reacts with Cu_4_(SO_4_)(OH)_6_ to produce a CuS precipitate. When the ratio of Cu^2+^ to S^2−^ ion is 1:1, CuS is precipitated in the solution, and the reaction ends. When the ratio of Cu^2+^ to S^2−^ is greater than 1:1 (Cu(CH_3_COO)_2_·H_2_O:TAA = 1.75:1, 1.8:1, 2:1), S^2−^ in the solution will reduce Cu^2+^, according to the amount of Cu^2+,^ and will eventually produce different copper sulfides Cu_7_S_4_, Cu_9_S_5_, Cu_7.2_S_4,_ and Cu_2_S. The specific equations (Equations (3–7)) are as follows:(2)$${\mathrm{CH}}_{3}{\mathrm{CSNH}}_{2}+3{\mathrm{OH}}^{-}\overset{\mathrm{heating}}{\to }{\mathrm{CH}}_{3}{\mathrm{COO}}^{-}+{\mathrm{NH}}_{3}+S^{2-}+H_{2}O$$
(3)$$S^{2-}+{\mathrm{Cu}}^{2+}\overset{\mathrm{heating}}{\to }\mathrm{CuS}↓$$
(4)$$\;4S^{2-}+7{\mathrm{Cu}}^{2+}\overset{\mathrm{heating}}{\to }{\mathrm{Cu}}_{7}S_{4}↓$$
(5)$$5S^{2-}+9{\mathrm{Cu}}^{2+}\overset{\mathrm{heating}}{\to }{\mathrm{Cu}}_{9}S_{5}↓$$
(6)$$4S^{2-}+7.2{\mathrm{Cu}}^{2+}\overset{\mathrm{heating}}{\to }{\mathrm{Cu}}_{7.2}S_{4}↓$$
(7)$$S^{2-}+2{\mathrm{Cu}}^{2+}\overset{\mathrm{heating}}{\to }{\mathrm{Cu}}_{2}S↓$$

In addition, as can be seen from the enlarged figure in Figure 2a, with the hydrothermal reaction, the crystal nuclei of copper sulfide grow into uniform nanoparticles. Then, due to the low viscosity of the solvent, the nanoparticles become close to agglomeration [11]. They come together, fuse at the boundary, and then eventually form aggregated particles of copper sulfide.

#### 2.1.3. Electrochemical Test Analysis

Figure 3a shows cyclic voltammetry (CV) curves of CuS, Cu_7_S_4_, Cu_9_S_5_, Cu_7.2_S_4,_ and Cu_2_S. (A three-electrode test method was adopted using a self-made electrode working electrode, a calomel electrode as a reference electrode, and a platinum electrode as a reverse electrode, a 2 mol/L KOH solution as an electrolyte solution, a scanning rate 5 mV/s, and a scanning potential range 0–0.6 V.) It can be seen from Figure 3a that the CV curves of the five copper sulfides all have obvious REDOX peaks, indicating that the electrochemical reaction of copper sulfides in circulation has a pseudocapacitive reaction. In addition, the oxidation and reduction peaks of the four copper sulfides are relatively symmetrical, indicating that the reaction reversibility of the four copper sulfides is good. However, the oxidation peak of cuprous sulfide is significantly higher than its reduction peak, indicating that the reversibility of cuprous sulfide is poor in the process of charge and discharge. By comparing the area enclosed by the CV curve, cuprous sulfide is significantly larger than other copper sulfides, which indicates that its energy storage performance is superior to other copper sulfides.

The material capacitance contribution rate b can be calculated by Equation (8) (b = 0.5, the electrode material behaves as battery property; b∈(0.5–1), and the electrode material shows the properties of battery and pseudocapacitance (b ≥ 1, and the electrode material exhibits pseudocapacitance properties). In Equation (8), where i is the peak current in the CV figure, v is the scanning rate, and a is a constant. Equations (8) and (9) is transformed by formula transformation. Then, a binary system of first order equations was established through multiple sets of data to eliminate the constant a; Equation (10) was then obtained. According to Equation (10), six different $\log\left(i_{2}/i_{1}\right)$ and $\log\left(v_{2}/v_{1}\right)$ groups were calculated by combining the data with the peak current i of scanning rate v of 5, 10, 20, and 50 mV/s, and b was obtained by fitting the data of six groups. According to the fitting data in Figure 3b, the capacitance contribution ratio b of CuS, Cu_7_S_4_, Cu_9_S_5,_ and Cu_2_S is 1.21, 1.08, 1.11, and 1.1, respectively. The values are all greater than 1, so copper sulfide as a supercapacitor material shows a pseudocapacitance property. This result corresponds to the CV curve in Figure 3a.
(8)$$i={\mathrm{av}\;}^{b}$$
(9)logi=blogv+loga
(10)$$b=\log\left(\frac{i_{2}}{i_{1}}\right)/\log\left(\frac{v_{2}}{v_{1}}\right)$$

According to the CV figure in Figure 3a, the CV curves of Cu_2_S has the largest area at the scanning rate of 5 mV/s, indicating that Cu_2_S has the best energy storage capacity. In order to better explore the electrochemical properties of the material, the constant current charge-discharge curve of the material was tested, and the specific capacitance of the material was calculated by Equation (11) [12]. (I is the discharge current; Δt is the charging and discharging time; Δv is put the potential range of filling; m is the quality of the active substance).
(11)$$\mathrm{Cm}=I\Delta t/\left(m\;\Delta v\right)$$

Figure 4b is the GCD diagram of the material at a current density of 1 A/g. The specific capacitances of CuS, Cu_7_S_4_, Cu_7.2_S_4,_ and Cu_2_S were calculated as 43.2, 86.3, 185.4, and 206.9 F/g. In addition, Cu_9_S_5_ has the same Cu–S content ratio as Cu_7.2_S_4_, both of which are 1.8:1. Like Cu_7.2_S_4_, Cu_9_S_5_ has a specific capacitance of 154 F/g, which is between Cu_7_S_4_ and Cu_2_S. It is found that with the increase of copper content in copper sulfide, the specific capacitance of the material increases, which also means that the energy storage capacity of the material is better. The results are in good agreement with the CV test. By observing the symmetry of GCD curves of five kinds of copper sulfides, different copper sulfides have excellent symmetry, which indicates that the coulomb efficiency of the materials is high. Equations (12) and (13) are the energy storage mechanisms of copper sulfide. According to the equations, the main reason for the energy storage mechanism of copper sulfide is the reversible pseudocapacitance reaction between copper sulfide and OH^−^ in the electrolyte, which can be explained as the continuous increase of copper valence. From this, the energy storage mechanism of the other copper sulfide can be deduced, as shown in Equation (14). According to the equations, with the increase of Cu element in CuS, Cu_7_S_4_, Cu_9_S_5_, Cu_7.2_S_4,_ and Cu_2_S, as well as the increase of x in Cu_x_S, more hydroxide can be reacted in the material, and the specific capacity of the material increases accordingly. In addition, according to the capacitance contribution rate b, copper sulfide exhibits pseudocapacitive properties when storing energy, which means that its ability to store charge completely comes from the reversible REDOX reaction of the electrode material. Therefore, with the increase of the Cu element in copper sulfide in Figure 4a, the REDOX pair provided by the material will increase accordingly, thus enabling the material to store more capacity, and its energy storage capacity will improve.
(12)$$\mathrm{CuS}+{\mathrm{OH}}^{-}↔\mathrm{CuSOH}+e^{-}$$
(13)$$\mathrm{CuSOH}+{\mathrm{OH}}^{-}↔\mathrm{CuSO}+H_{2}O+e^{-}$$
(14)$${\mathrm{Cu}}_{x}S+\left(4x-2\right){\mathrm{OH}}^{-}↔{\mathrm{Cu}}_{x}{\mathrm{SO}}_{\left(2x-1\right)}+\left(2x-1\right)H_{2}O+\left(4x-2\right)e^{-}$$

The impedance of the material and strength of the charge-electron transport capacity can be analyzed by the AC impedance test. The test curve can be shown in the semi-circular diameter of the high-frequency region, which represents the charge-transfer resistance (Rct), and the low-frequency oblique line represents the diffusion resistance (Rw) of the electrode material, respectively. The electron transport speed in the reaction process is related with the Rct, and the slope of the oblique line is related with the Rw. The higher the slope of the oblique line, the lower the Rw. The ionic impedance of the electrolyte, the internal resistance of the electrode itself, and the interfacial impedance between the active material and the nickel foam collector can be shown as the intercept of the curve on the real axis, which represents the resistance (Rs).

Figure 4c shows the electrochemical impedance spectroscopy (EIS) of different copper sulfides. It can be seen from the figure that the high-frequency curves of the EIS diagrams of the five materials are almost ignored, indicating that the internal resistance of copper sulfide is very small when it is used as the material of supercapacitor, which proved that it has good energy storage performance. The low-frequency EIS curves of CuS, Cu_7_S_4_, Cu_9_S_5_, Cu_7.2_S_4,_ and Cu_2_S were analyzed. In this section, with the increase of the copper stoichiometry in the copper sulfide, the slope of the straight line in the low-frequency region gradually increases, indicating that the resistance of the interface charge transfer of the material is smaller and the diffusion resistance of the electrolyte of the material is lower. This also proves that its electrochemical ability is better.

Table 1 shows the specific capacitance and capacitance retention of different copper sulfides. According to the table, the retention rate of Cu_9_S_5_ reached 98.5%, which was higher than that of the other four materials. In addition, the capacitance retention of Cu_7_S_4_ and Cu_2_S is 96.3% and 93.7%, which also show excellent capacitance retention. The retention rate of Cu_7.2_S_4_ was lower than that of the first three copper sulfides, which maintained at 80.3%. However, the capacitance retention of CuS is the lowest, only 63.4%. This may be related to the stability of the material. According to the CRC Handbook of Chemistry and Physics of 2014–2015 edition, the standard molar entropy of copper sulfide is 66.5 J/mol K, and that of cuprous sulfide is 120.9 J/mol K. This indicates that the structural stability of Cu_2_S is better than CuS. Therefore, as a pseudocapacitive electrode, when the pseudocapacitive reaction occurs with the increase of current density, the resistance to a large current is stronger due to improved stability, so the capacitance retention rate is higher [13,14,15,16,17].

Figure 5 is the GCD cycle diagram of different copper sulfides at 10 A/g current density. It can be observed from Figure 5a–e that the cycle curve keeps an upward trend from 0 to 200 cycles. It shows that the specific capacitance of the material is increasing, and that the energy storage is increasing. The main reason is that the morphology of copper sulfide is agglomerate. With the continuous charge and discharge, the material continuously generates pseudocapacitive reactions. Then, the piles of copper sulfide spread out, increasing the specific surface area. Finally, the number of copper elements involved in the reaction continues to increase, thus increasing the charge stored in the material [18].

It can be seen from the analysis of Figure 5 that Cu_2_S has the best cycle stability, and its capacity increases the most. It increased by 255.8% to 547.9 F/g. Secondly, the cyclic stability of Cu_9_S_5_ was excellent and remained stable until 2000 cycles, and the specific capacity increased by 130.3%. The cyclic stability of Cu_7_S_4_ is general. After rising, it stabilizes for a period of time, and then decreases with a small increase. The cycle stability of CuS is poor, and it begins to decline after 200 cycles without stable interval.

Finally, the cycle stability of Cu_7.2_S_4_ is the worst, with no rising period and no stable region. After 1000 cycles, the specific capacity is only 44.5%. The cyclic stability of materials is related to the structural stability of materials. As the material is recycled, the original structure will continue to collapse. As a result, the number of materials involved in the reaction will continue to decrease, resulting in a decrease in the specific capacity of the material. With a high and non-integral Cu–S ratio, the crystal structure of Cu_7.2_S_4_ may be more unstable, which resulted it in being infected the deepest and collapsing the most easily while the material was recycled.

### 2.2. Effect of Morphology of Cuprous Sulfide on Electrochemical Properties of Materials

#### 2.2.1. XRD Analysis

Figure 6 is the XRD analysis of Cu_2_S(I), Cu_2_S(II), and Cu_2_S(III). It can be seen from the figure that the strength peaks of cuprous sulfide with three morphs are completely corresponding to the standard card of cuprous sulfide. The crystal faces of (412), (204), (630), (106), and (112) of cuprous sulfide at the 2θ angles are 36.1°, 37.3°, 45.8°, 48.3°, and 53.5°. Therefore, it can be determined that the synthesized substances Cu_2_S(I), Cu_2_S(II), and Cu_2_S(III) are pure cuprous sulfide by adjusting the solvent of the solvothermal reaction and the subsequent annealing treatment of the hydrothermal products.

#### 2.2.2. SEM Analysis

Figure 7 shows the SEM images of Cu_2_S(I), Cu_2_S(II), and Cu_2_S(III). Through SEM images, we can observe that the cuprous sulfide prepared by the three synthetic methods has different morphologies. The materials shown in Figure 7a,b are also prepared by hydrothermal synthesis. The aggregate size of the synthesized large particles of cuprous sulfide is concentrated between 20 μm and 55 μm. Cuprous sulfide with a porous spherical structure, as shown in Figure 7c,d, were successfully prepared by changing the reaction solvent to 99% purity ethanol solution. The main reason is that the ethanol is weak acidic, which makes the S^2−^ and H^+^ generated by the decomposition of thioacetamide generate H_2_S, thus reducing the nucleation rate of cuprous sulfide and affecting the final morphology. In addition, compared with distilled water, ethanol as a solvent has lower viscosity and lower surface tension. Therefore, cuprous sulfide crystals are subjected to less external force during the growth process so that the crystals can expand more outwardly. In turn, the agglomeration is reduced and larger pores are formed, resulting in a porous morphology.

Figure 7e,f are the scanning images of Cu_2_S(III) prepared by annealing at 600 °C for 3 h with micropore structure cuprous sulfide. As can be seen from Figure 7d, the porous cuprous sulfide is formed by the fusion and combination of many nanoparticles. Since the subsequent annealing temperature of 600 °C is higher than the recrystallization temperature of cuprous sulfide, the junction of porous structure grains gradually melts under the action of high temperature, and the cuprous sulfide nanoparticles are separated. As the temperature increases, the particles gradually homogenize and eventually form nanoparticles cuprous sulfide. As can be seen from Figure 7f, the size of cuprous sulfide nanoparticles is concentrated at 500 nm.

#### 2.2.3. Electrochemical Test Analysis

It can be seen from Figure 8 that cuprous sulfide with three morphologies all have obvious REDOX peaks. According to the calculation of the capacitance contribution rate in the last work, there is no double electric layer effect in cuprous sulfide energy storage, indicating that the Faraday reaction occurs in the process of energy storage. In addition, by comparing the symmetry of the REDOX peak of the three materials, it can be seen that the oxidation peak height of cuprous sulfide with the three morphologies is obviously higher than its reduction peak height. It shows that the reversibility of materials is related to the properties of materials and has nothing to do with the morphology of materials. By comparing the areas enclosed by the CV curves of cuprous sulfide in a different morphology, it can be seen that the area enclosed by the agglomerated cuprous sulfide is the largest, followed by the CV area in the microporous condition, and the smallest is the nanoparticle. It is proved that when the morphology is agglomerate, cuprous sulfide has more excellent energy storage capacity, while nanoparticles have the worst energy storage capacity. In addition, the agglomerated cuprous sulfide has a higher current intensity, indicating that it has a faster REDOX kinetics during charge and discharge [19,20].

As shown in Figure 9, the influence of morphology on the energy storage performance of materials was explored by conducting a constant current charge and discharge at different current densities for cuprous sulfide with different morphologies. Figure 9a,b show the charge-discharge curves of agglomerated and microporous cuprous sulfide at the current density of 1, 3, 5, 10 A/g. It can be seen from the figure that the symmetry of the curve is improved, indicating that the coulomb efficiency of both is improved. Figure 9c shows the charge–discharge curve of cuprous sulfide nanoparticles. It can be seen from the figure that at the current density of 1 A/g, the charging time of the material is significantly longer than the discharging time, indicating that the coulomb efficiency of the material is poor at this current density. The reason is that the surface of the nanoparticles is flat, which makes it difficult for OH^−^ to attach during pseudocapacitance reaction. The separation and embedding of OH^−^ are affected, which makes the coulombic efficiency of the material poor. However, by increasing the current density, the symmetry of the material becomes better, indicating that increasing the current is conducive to the oxidation process of nano cuprous sulfide, thus obtaining excellent coulomb efficiency [21,22].

The specific capacitance of the material can be calculated by the GCD curve of the material. Table 2 shows that the specific capacitance of cuprous sulfide with agglomerated, microporous, and nanoparticle morphology at the current density of 1 A/g is 206.9, 169.2 and 147.2 F/g. The specific capacitance at the current density of 10 A/g is 193.8, 159.2, and 140.6 F/g, respectively. The capacitance retention rates of the three materials were 93.7%, 94.1%, and 97%, respectively, and the best capacitance retention rate was obtained under the morphology of nano particles. The important reason is that compared with the other two kinds of morphologies, the nanoparticle morphology has better structural stability and stronger resistance when facing large current, so the retention rate is the best.

#### 2.2.4. Specific Surface Analysis

According to Figure 10, it can be known that intensive reunion Cu_2_S and nanoparticles Cu_2_S has no obvious hysteresis ring, indicating that there is basically no pore structure. The microporous structure has a hysteretic ring, but the ring is not regular, indicating that the pore size distribution is uneven. The specific surface area of the agglomerated cuprous sulfide was 11 cm^2^/g. The specific surface area of the microporous cuprous sulfide is 3.22 cm^2^/g. The cuprous sulfide of nanoparticles was only 0.11 cm^2^/g. It is found that the specific surface area corresponds to the specific capacitance of the material. It shows that the energy storage capacity of the material increases with the increase of the specific surface area. The main reason is that in the process of pseudocapacitive reaction, a large specific surface area can provide more active sites so that more materials can participate in the reaction and, thus, more energy can be stored [23].

Figure 11a shows the AC impedance curves of agglomerated, microporous, and nanoparticular cuprous sulfide. It can be seen from the figure that the semicircular radius of the high-frequency zone curve of the agglomerated cuprous sulfide is the smallest, indicating that the electrical conductivity of the agglomerated cuprous sulfide is the best under this topography. The cuprous sulfide of nanoparticles has the largest radius, indicating that its electrical conductivity is worse than that of other morphology [24]. On the whole, although the electrical conductivity of materials with different morphologies is different, the curves in the high-frequency region of cuprous sulfide materials with three morphologies are almost ignored. It shows that the internal resistance of cuprous sulfide with different morphology is very small when used as supercapacitor materials.

The EIS curves of agglomerated, microporous, and nanoparticle Cu_2_S were analyzed at a low frequency, and it was found that the slope of the line in the microporous Cu_2_S low-frequency region was the highest. The results indicate that the resistance of interfacial charge transfer and the diffusion resistance of the electrolyte are lower when the material is used as the electrode. The reason is that the microporous Cu_2_S has larger pores inside, which makes the diffusion channel of electrolyte wider than the other two morphologies. As a result, the diffusion resistance of ions is smaller, which is more conducive to the diffusion of ions, resulting in a smaller interfacial charge transfer resistance and a lower diffusion resistance of the electrolyte [25].

Figure 11b shows the influence of morphology on the cyclic stability of cuprous sulfide materials. As shown in Figure 11, the specific capacitance of agglomerated particle morphology at the beginning of the cycle is 193.8 F/g. Then, the capacity continues to rise to the maximum capacity of 547.9 F/g to 2300 laps. By Lap 3000, the remaining specific capacity was only 529.2 F/g. The square dot curve is the cycle graph of cuprous sulfide nanoparticles. As can be seen from the figure, the specific capacity of the material continues to rise from 140.6 F/g at the beginning to 227 F/g after 200 cycles during 0–3000 cycles. Then, it grows steadily to 3000 laps and stays steady. It briefly increases to a maximum of 340.2 F/g at 3400 laps. By 4000 cycles, the specific capacity was only 336.8 F/g. The triangulation curve shows the specific capacitance of the microporous cuprous sulfide during 3000 charge-discharge cycles. As shown in the figure, the microporous cuprous sulfide increases briefly from 159.2 F/g to 175.4 F/g. Then, it keeps going down. After 3000 cycles, only 129.6 F/g is left.

From the above analysis, it can be seen that both agglomerated cuprous sulfide and nanoparticle cuprous sulfide have good cycling stability, while the cycling stability of microporous cuprous sulfide is poor. The reason is that the stability of microporous cuprous sulfide is worse than the other structure. After repeated circulation, the microporous structure collapses, eventually forming a blockage. As a result, the specific surface area of the material decreases, and the active substances involved in the reaction decrease [26]. Finally, the specific capacity of the material continues to decline. The stability of cuprous sulfide particles is higher. Through a continuous reversible reaction, the active substances on the surface are constantly separated, thus making the specific surface area increase, making the specific capacity of the material increase. When the specific surface area increases to the limit, the specific capacity of the material begins to decline due to the cyclic loss of the material.

## 3. Materials and Methods

### 3.1. Chemicals and Materials

The experimental materials of thioacetamide (TAA), monohydrate copper acetate, sodium hydroxide, polyvinylidene fluoride (PVDF), *N*-methylpyrrolidone (NMP), acetylene black, distilled water, and 99.9% alcohol are analytical pure from Aladdin.

### 3.2. Synthesis of Different Copper Sulfides

By hydrothermal synthesis [27,28], 70 mL of distilled water was added into the reactor. Then, 1 mmol of copper acetate monohydrate was added and stirred until it was completely dissolved. In addition, 1 mmol of TAA was added, and the solution immediately generated flocculent precipitation. Then, the pH value of the solution was adjusted to alkaline by adding 3.5 mmol of sodium hydroxide. Then, the solution was transferred to a drying oven and kept at 180 °C for 24 h to get black precipitation. Finally, Cu_x_S_y_ (I, CuS) was obtained by freeze-drying after cleaning it to neutral. Under the same method, Cu_x_S_y_ (III, Cu_9_S_5_) and Cu_x_S_y_ (V, Cu_2_S) were obtained by adjusting the amount of monohydrate copper acetate to 1.8 mmol and 2 mmol. Cu_x_S_y_ (IV, Cu_7.2_S_4_) was obtained by changing the pH value of 1.8 mol solution with potassium hydroxide. The ratio of copper acetate monohydrate to TAA was 1.75:1, and other reaction conditions remained the same, so Cu_x_S_y_ (II, Cu_7_S_4_) was obtained.

### 3.3. Morphology Control of Cuprous Sulfide

Repeat 2.2 to obtain Cu_2_S (I). We changed 70 mL of distilled water in the reactor into 70 mL of alcohol, while 2 mmol of monohydrate copper acetate and 1 mmol of TAA were successively added. Then, Cu_2_S (II) was prepared by vacuum drying oven with 180 °C of heat preservation for 24 h. The Cu_2_S (II) was transferred to the sintering furnace and annealed in argon atmosphere at 600 °C for 3 h to obtain Cu_2_S (III).

### 3.4. Preparation of Test Electrodes 

Common electrochemical workstation was used for electrode test and electrochemical test. The reference electrode was calomel electrode, the reverse electrode was platinum electrode, and the electrolyte solution was 2 mol/L of KOH in the test. We added 2 mg of PVDF and an appropriate amount of NMP into a 5 mL flask and electromagnetically stirred for 1.5 h. Then, 2 mg of acetylene black was added and stirred for 0.5 h. Active substances (Cu_x_S_y_ (I, II, III, IV and V) and Cu_2_S (I, II and III)) were added for 16 mg, sealed, and stirred for 12 h to get the solution. We evenly dropped 0.3 mL of solution on the round sheet nickel foam wafer with a diameter of 10 mm, and then ventilated and dried it for 8 h. It was then vacuum dried for 12 h. Finally, the small wafer was connected with the pole ear at a pressure of 10 Mpa, and the test electrode was prepared by vacuum drying for 12 h.

## 4. Conclusions

Different stoichiometric coefficient copper sulfides, such as CuS, Cu_7_S_4_, Cu_9_S_5_, Cu_7.2_S_4,_ and Cu_2_S with uniform morphology, were prepared by a hydrothermal method. Electrochemical tests show that different copper sulfides exhibit pseudocapacitance properties when used as electrode materials for supercapacitors. In addition, the specific capacitance of copper sulfide increases with the increase of copper stoichiometric coefficient to sulfur. The main reason is that the increase of active sites makes more OH^−^ participate in the pseudocapacitive reaction. It can be seen from the cyclic test at the current density of 10 A/g that cuprous sulfide has the best circulability. After activation, the specific capacitance can reach 547.9 F/g. Cu_9_S_5_ was followed by Cu_7_S_4_, CuS was average, and Cu_7.2_S_4_ was the worst. The agglomerated cuprous sulfide was successfully synthesized by the hydrothermal method. The microporous cuprous sulfide was successfully prepared by changing the reaction solvent. The nanometer cuprous sulfide was obtained by annealing microporous cuprous sulfide. Electrochemical tests show that the morphology has little effect on the pseudocapacity and impedance of cuprous sulfide. However, the specific capacitance is proportional to the specific surface area of different morphologies. In addition, the circulation of microporous cuprous sulfide is the worst. The reason is that its structure stability is poor; with the progress of the cycle, the structure of the material is destroyed, resulting in the constant decline of specific capacitance.

## Figures and Tables

**Figure 1 molecules-28-02487-f001:**
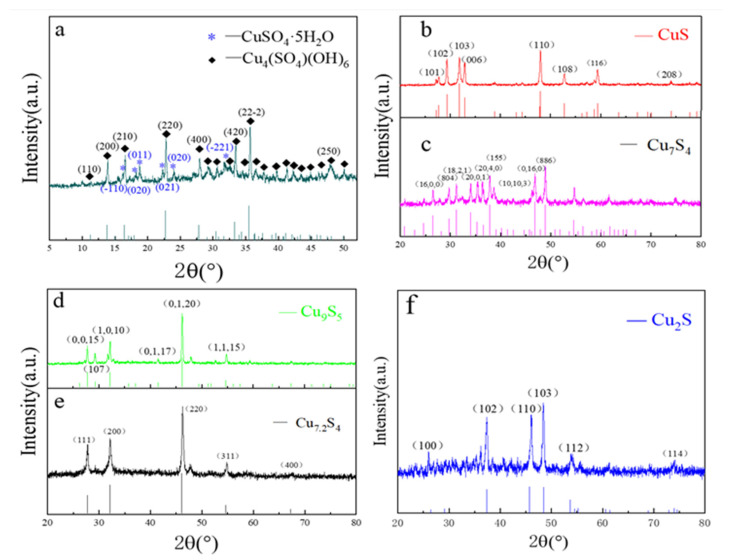
(**a**) is the XRD patterns figure of intermediates prepared by Cu(CH_3_COO)_2_·H_2_O and TAA; (**b**–**f**) are the XRD patterns figures of CuS, Cu_7_S_4_, Cu_9_S_5_, Cu_7.2_S_4,_ and Cu_2_S.

**Figure 2 molecules-28-02487-f002:**
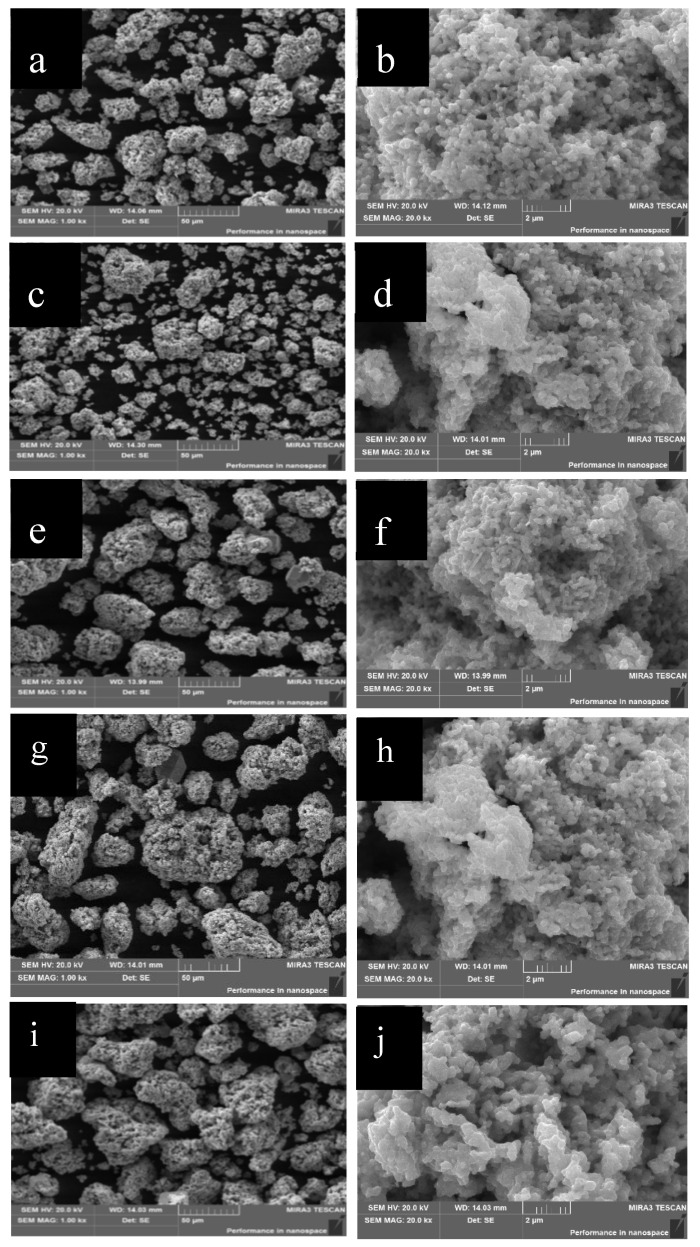
The morphology SEM images of different copper sulfides (**a**,**c**,**e**,**g**,**i**) (CuS, Cu_7_S_4_, Cu_9_S_5_, Cu_7.2_S_4,_ and Cu_2_S), whose particle diameter is 55 μm; Figures (**b**,**d**,**f**,**h**,**j**) are enlarged image part of figures (**a**,**c**,**e**,**g**,**i**).

**Figure 3 molecules-28-02487-f003:**
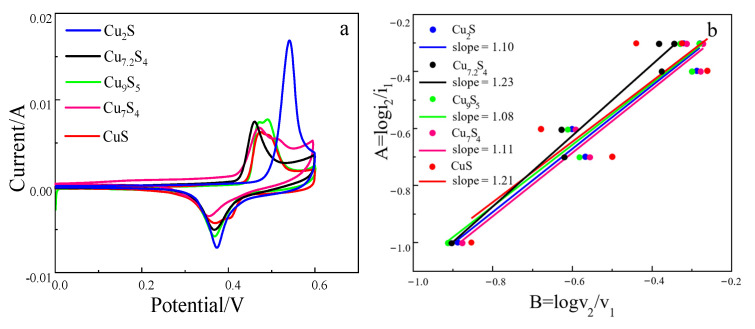
(**a**) is the CV curves of a CuS, Cu_7_S_4_, Cu_9_S_5_, Cu_7.2_S_4,_ and Cu_2_S at scanning rate of 5 mV/s; (**b**) is the capacitance contribution b of different copper sulfides.

**Figure 4 molecules-28-02487-f004:**
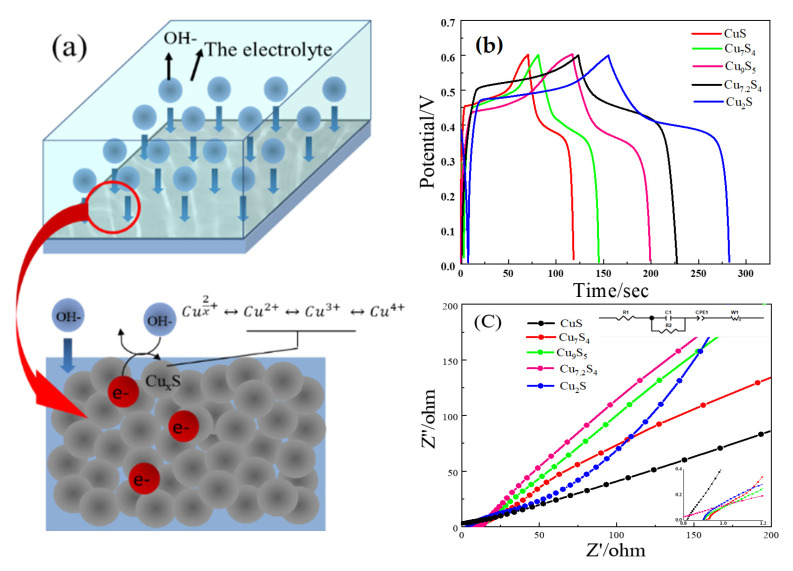
The schematic diagram of (**a**) energy storage of copper sulfide; the GCD diagrams of (**b**) different copper sulfides (CuS, Cu_9_S_5_, Cu_7_S_4_, Cu_7.2_S_4,_ and Cu_2_S) at 1 A/g current density; the electrochemical impedance spectroscopy (EIS) diagrams and equivalent circuit of (**c**) different copper sulfides.

**Figure 5 molecules-28-02487-f005:**
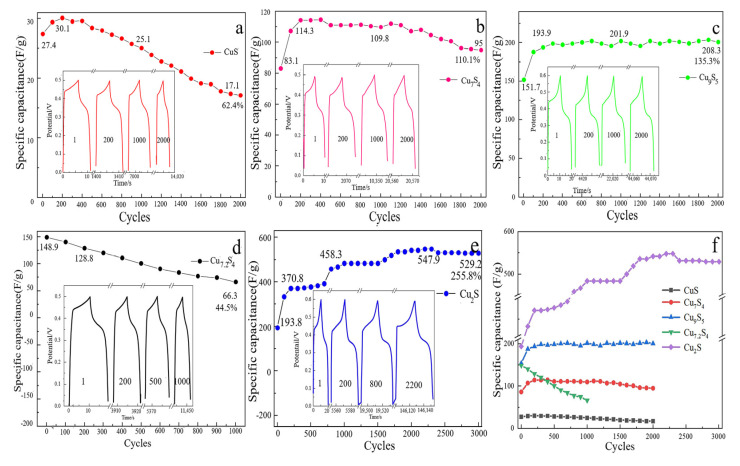
(**a**) to (**e**) are different cyclic properties of copper sulfides at 10 A/g current density: CuS (**a**), Cu_7_S_4_ (**b**), Cu_9_S_5_ (**c**), Cu_7.2_S_4_ (**d**), Cu_2_S (**e**); a synthetic contrast of cyclic properties of (**f**) five kinds of copper sulfides.

**Figure 6 molecules-28-02487-f006:**
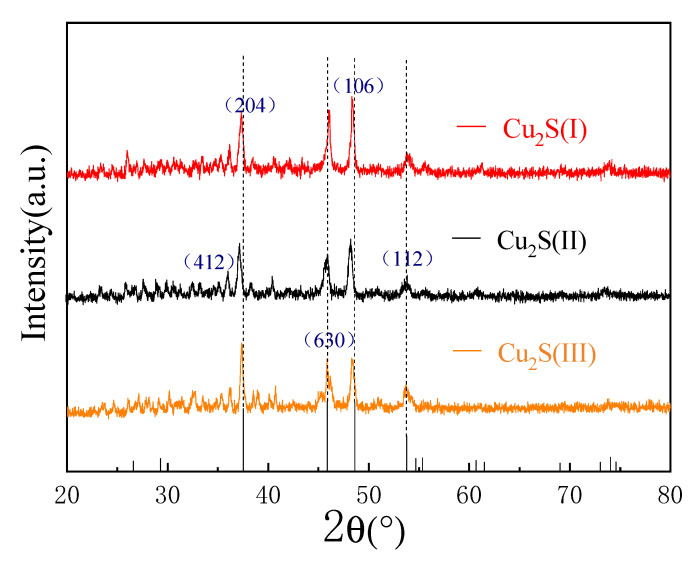
XRD patterns of cuprous sulfide with three different morphologies: Cu_2_S(I), Cu_2_S(II), and Cu_2_S(III).

**Figure 7 molecules-28-02487-f007:**
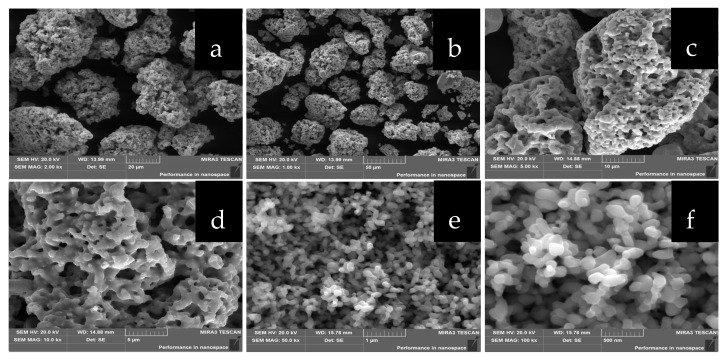
SEM images of cuprous sulfide with different morphologies: the scanning images of (**a**,**b**) Cu_2_S(I); the scanning images of (**c**,**d**) Cu_2_S(II); the scanning images of (**e**,**f**) Cu_2_S(III).

**Figure 8 molecules-28-02487-f008:**
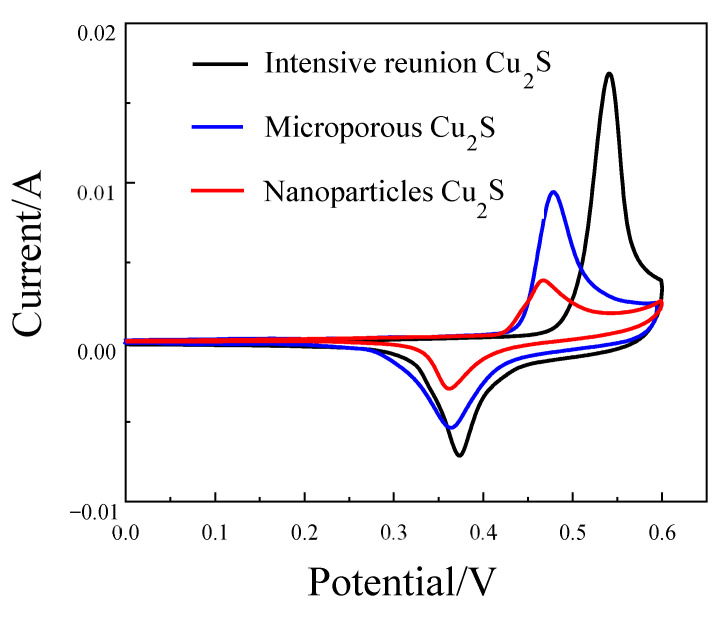
The cyclic voltammetry diagram curves of intensive reunion Cu_2_S, microporous Cu_2_S, and nanoparticles Cu_2_S.

**Figure 9 molecules-28-02487-f009:**
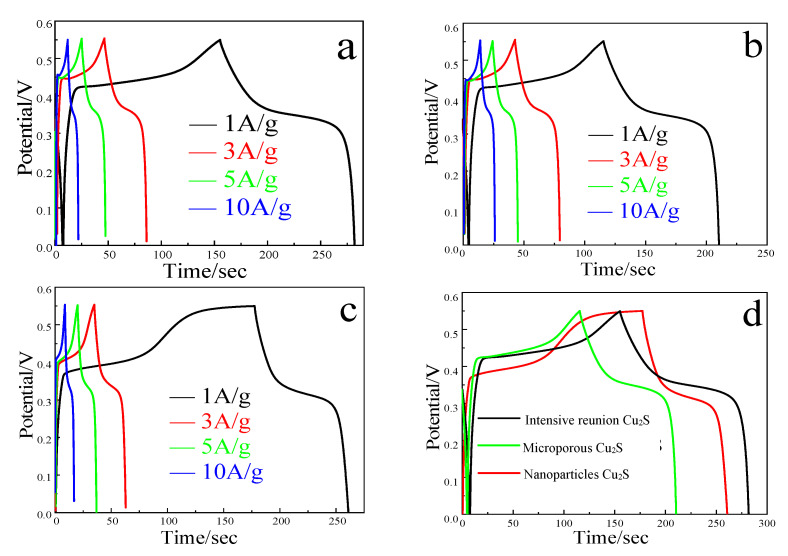
The GCD curves of (**a**–**c**) intensive reunion, microporous and nanoparticles three morphologies of Cu_2_S; capacitance retention of (**d**) three different morphologies.

**Figure 10 molecules-28-02487-f010:**
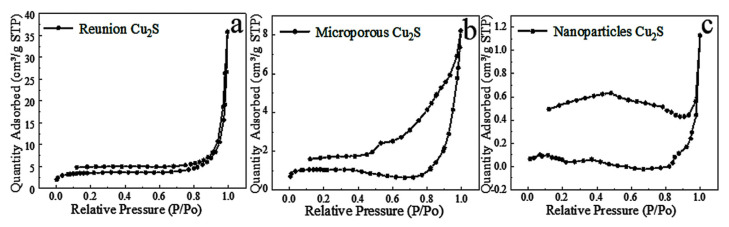
Nitrogen adsorption and desorption isotherms of Cu_2_S with different morphologies.

**Figure 11 molecules-28-02487-f011:**
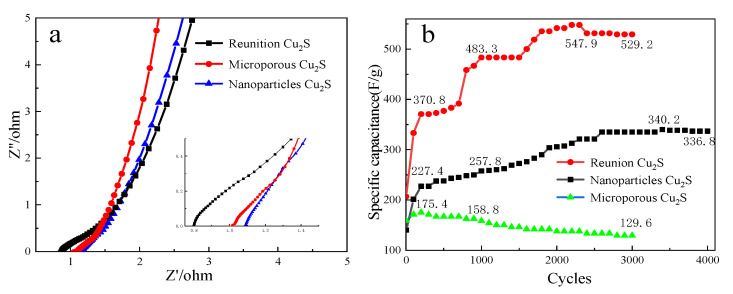
Different impedance curves of (**a**) three different morphologies of Cu_2_S; different cyclic stability of (**b**) three different morphologies of Cu_2_S.

**Table 1 molecules-28-02487-t001:** The capacitance capability of different materials with the same morphology (CuS, Cu_9_S_5_, Cu_7_S_4_, Cu_7.2_S_4,_ and Cu_2_S) Cu–S ratio at 1 A/g and 10 A/g current density.

Material	Cu–S Ratio	C(1 A/g)	C(10 A/g)	Retention
CuS	1:1	43.2 F/g	27.4 F/g	63.4%
Cu_7_S_4_	1.75:1	86.3 F/g	83.1 F/g	96.3%
Cu_9_S_5_	1.8:1	154 F/g	151.7 F/g	98.5%
Cu_7.2_S_4_	1.8:1	185.4 F/g	148.9 F/g	80.3%
Cu_2_S	2:1	206.9 F/g	193.8 F/g	93.7%

**Table 2 molecules-28-02487-t002:** Specific capacitance and capacitance capability of cuprous sulfide with different morphologies.

Material	C(1 A/g)	C(10 A/g)	Retention
Intensive reunion Cu_2_S	206.9 F/g	193.8 F/g	93.7%
Microporous Cu_2_S	169.2 F/g	159.2 F/g	94.1%
Nanoparticles Cu_2_S	147.2 F/g	140.6 F/g	97%

## Data Availability

The data that support the findings of this work are available from the corresponding author upon reasonable request.

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
