# Peer review of "The Effect of Copper Sulfide Stoichiometric Coefficient and Morphology on Electrochemical Performance"

_molecules, 2023, doi:10.3390/molecules28062487_

Round 1
Reviewer 1 Report
Report on the manuscript “The effect of copper sulfide stoichiometric coefficient and morphology on electrochemical performance”, Y Ding et al.
Ref. 2246288.
General comments:
The manuscript describes the synthesis, characterization and energy storage performance of a series of copper suifides (CuS, Cu7S4, Cu9S5, Cu7.2S4 and Cu2S) prepared by hydrotermal method. This is a conventional electroanalytical manuscript describing a potentially interesting sensor but offering significant weaknesses. Accordingly, major revision is recommended based on the following considerations.
General remarks:
I) The manuscript is uneasy to follow due to the peculiar distribution of contents. The synthesis is delayed to the end of the Results and discussion section and, in this, the description of XRD data starts the sub-section 2.1 “Energy storage …”. Logically, a “Characterization” sub-section should be included.
II) Following the same line of reasoning, the figure captions for the voltammograms in Figures 3 and 8, as well as EIS spectra in Figure 11, and all GCD curves do not provide the electrolyte.
III) The electrochemical processes observed in the CVs in Figs. 3 and 8 should be described.
IV) The analysis of EIS data is simplistic. Experimental impedance spectra should be fitted to a suitable equivalent circuit.
V) As is unfortunately frequent in recent literature, the authors do not perform a judicious analysis of uncertainties based on the performance of replicate experiments. This is particularly relevant with regard of data in Figure 3b, where, given the high data dispersion, there is no possibility of establishing a reasonable discrimination between the distinct copper sulfides. In short: the capacitance contribution ratios (1.21, …1.1) fall probably within the range of experimental error; i.e., there are no significant differences.
VI) This problem is aggravated because, due to the electrode preparation procedure described at the end of the manuscript, there is no possibility to ensure that the net amount of copper sulfide was the same in all cases. This means that the comparison between the different copper sulfides relative to mass is uncertain.
VII) As indicated in #V, the absence of a judicious analysis of errors is reflected in the expression of several parameters (specific capacitances in particular) with an unrealistic number of significant figures (Table 2, main text) if replicate experiments are carried out. This is a frequent problem in literature that should be corrected. The authors should be aware that writing, for instance, 529.2 F/g means that this quantity is known with a relative uncertainty of only 0.019 % (!!).
Reviewer 2 Report
The authors prepared five kinds of copper sulfides by hydrothermal method and investigated the stoichiometric coefficient and morphology on the electrochemical performance in supercapacitors. The obtained electrochemical performance is reasonable, and the result is well analyzed. However, some of the issues need to be addressed before publishing the manuscript in Molecules. Hence, I recommend that minor revisions are required to meet the journal requirement.
1. In Figure 1d and 1e, there are obvious small peaks near 2θ≈48°. What substances do these peaks correspond to?
2. Please check the format of all references according to the journal requirements.
3. In the electrochemical part, the formula of cyclic voltammetry for calculating the specific capacitance is missing.
4. In the synthesis of Cu7.2S4, pH value is 1.8 mol solution of potassium hydroxide while in case of CuS, Cu2S, Cu9S5, pH value is 3.5 mmol of sodium hydroxide. Why?
5. The manuscript has some shortcomings in grammar, typing errors and structure, which should be corrected in the revised manuscript.
Round 2
Reviewer 1 Report
The manuscript can be published in its current version.